# Intestine-to-neuronal signaling alters risk-taking behaviors in food-deprived *Caenorhabditis elegans*

Molly A. Matty[1]☯, Hiu E. Lau[1,2]☯, Jessica A. Haley[1,3], Anupama Singh[1,4], Ahana Chakraborty[1], Karina Kono[1], Kirthi C. Reddy[1], Malene Hansen[4], Sreekanth H. Chalasani[1]*

1 Molecular Neurobiology Laboratory, The Salk Institute for Biological Studies, La Jolla, California, United States of America, 2 Division of Biological Sciences, University of California, San Diego, La Jolla, California, United States of America, 3 Neurosciences Graduate Program, University of California, San Diego, La Jolla, California, United States of America, 4 Development, Aging and Regeneration Program, Sanford Burnham Prebys Medical Discovery Institute, La Jolla, California, United States of America

☯ These authors contributed equally to this work.
* schalasani@salk.edu

**Data Availability Statement:** All relevant data are within the manuscript and its Supporting information files.

## Abstract

Animals integrate changes in external and internal environments to generate behavior. While neural circuits detecting external cues have been mapped, less is known about how internal states like hunger are integrated into behavioral outputs. Here, we use the nematode *C. elegans* to examine how changes in internal nutritional status affect chemosensory behaviors. We show that acute food deprivation leads to a reversible decline in repellent, but not attractant, sensitivity. This behavioral change requires two conserved transcription factors MML-1 (MondoA) and HLH-30 (TFEB), both of which translocate from the intestinal nuclei to the cytoplasm during food deprivation. Next, we identify the insulin-like peptide INS-31 as a candidate ligand relaying food-status signals from the intestine to other tissues. Further, we show that neurons likely use the DAF-2 insulin receptor and AGE-1/PI-3 Kinase, but not DAF-16/FOXO to integrate these intestine-released peptides. Altogether, our study shows how internal food status signals are integrated by transcription factors and intestine-neuron signaling to generate flexible behaviors via the gut-brain axis.

## Author summary

We have all experienced behavioral changes when we are hungry—the pang in our stomach that can cause us to behave erratically. In particular, animals, including humans, are known to pursue more risky behaviors when they are hungry as compared to when they are well-fed. Here we explore the molecular details of this behavior in the invertebrate animal model *C. elegans*. We show that *C. elegans* displays reduced copper sensitivity when hungry. As copper is toxic and repellant to *C. elegans*, this reduced avoidance suggests that these animals employ riskier food search strategies when food-deprived. Moreover, we find that this hunger-induced behavioral change is reversible upon re-feeding and is not caused by an increased attraction to food or depletion of fat stores, but rather insulin

**Funding:** This work was supported by grants from The Rita Allen Foundation (http://ritaallen.org), The W.M. Keck Foundation (http://wmkeck.org), NIH R01 MH096881 (S.H.C.); NSF Postdoctoral Research Fellowship in Biology Grant No. 2011023 (M.A.M.); Glenn Foundation Fellowship (http://https://www.salk.edu/science/research-centers/glenn-center-for-research-on-aging/) (A.S.); NSF Socrates Award 742551 (H.E.L.); and NSF Graduate Research Fellowships (H.E.L. and J.A.H.). S.H.C. received salary support from grants from The Rita Allen Foundation, The W.M. Keck Foundation and NIH R01MH096881; M.A.M. received support from the NSF Postdoctoral fellowship; A.S. received support from the Glenn Foundation Fellowship; while H.E.L. and J.A.H. received support from the NSF fellowships. The funders had no role in study design, data collection and analysis, decision to publish, or preparation of the manuscript.

**Competing interests:** The authors have declared that no competing interests exist.

signaling between the intestine and neurons. We use genetic tools, microscopy, and behavioral tests to determine that this risky behavior involves a sensation of "lack of food" in the intestine, release of signaling molecules, and engagement with neurons. Our work highlights new and potentially evolutionarily conserved ways in which intestinal cells and neurons communicate and produce behavioral changes, highlighting the importance of the gut-brain-axis.

## Introduction

Animals evaluate their environment, integrating prior experiences and internal state information in order to optimize their behavior for maximum reward and threat avoidance [1]. Thus, changes in internal states play a critical role in adjusting the animal's responses to external stimuli [2,3]. One critical internal state is hunger, which has a profound effect on animal survival and elicits dramatic changes in food-seeking behaviors [2,4]. Multiple species, including humans, have been shown to alter their chemosensory behavior during periods of starvation [5–10]. Despite this, little is known about how the nervous system receives and interprets information about hunger status.

The nematode *Caenorhabditis elegans*, with just 302 neurons [11], and 20 intestinal cells [12], provides a unique opportunity for a high-resolution analysis of how the nervous system integrates internal signals. Previous studies have shown that, similar to mammals, *C. elegans* exhibits a number of behavioral, physiological, and metabolic changes in response to altered nutritional status. Hermaphroditic *C. elegans* retain eggs [13], are unlikely to mate with males [14], initiate altered foraging behaviors [15–17], and change their responses to environmental $CO_2$ [18], salt [19], and pheromones [20] upon food deprivation. Moreover, many molecules that signal hunger are conserved between *C. elegans* and vertebrates. For example, neuropeptide Y (NPY) signaling influences feeding behaviors in both nematodes and mammals [21–23]. Similar effects are also seen with insulin and dopamine signaling, which modify chemosensory neuronal activity in nematodes [24,25] and mammals [26–28] leading to changes in feeding behavior. While neuronal pathways responding to food-deprivation on the multiple-minute timescales have been mapped [17,29], those integrating these signals on the multiple hour timescales are poorly understood.

Here we used *C. elegans* to dissect the machinery required to integrate internal food signals and modify behaviors. We combined food deprivation over multiple hours with a behavioral assay that quantifies the animal's ability to respond to both toxic and food-related signals, mimicking a simplified ecologically relevant scenario. In this sensory integration assay, animals cross a toxic copper barrier (repellent) and chemotax towards a point source of a volatile food-associated odor, diacetyl (attractant) [30]. We show that animals that have been food-deprived for multiple hours have reduced sensitivity to the repellent and cross the copper barrier more readily than well-fed animals. Next, we show that two transcription factors change their localization patterns in the intestinal nuclei during multiple hours of food deprivation. We confirm a role for these transcription factors and identify the downstream peptide released by the intestine to relay the "lack of food" signal to other tissues. Finally, we show that neurons likely receive these intestine-released peptides. This allows food-deprived animals to use a higher risk strategy and search for food by reducing their avoidance to repellents.

## Results

### Acute food deprivation specifically alters repellent-driven behaviors

Animals simultaneously receive and interpret both attractant and repellent signals from their environment and use that information to generate appropriate behavioral responses. To mimic these ecological interactions, we employed a sensory integration assay in which animals must cross a repellent copper barrier ($CuSO_4$) towards a gradient of a volatile attractant, diacetyl [30] (described in Methods). The proportion of animals that cross the toxic copper barrier are counted and expressed as a chemotactic index (Fig 1A). We analyzed the behavior of well-fed, wildtype animals and found that only ~20% cross the copper barrier and locomote towards the spot of diacetyl (black bars, Fig 1B and 1C and S1 Movie). In contrast, animals food-deprived for at least 2 hrs were more likely to cross the copper barrier (blue bars, Fig 1B and S2 Movie) with a maximal effect at 3 hrs (Fig 1B and 1C). Next, we tested whether the food-deprivation effect was reversible. We food-deprived animals for 3 hrs and then returned them to food for increasing durations and analyzed animal behavior after the food experience. We found that 3-hour food-deprived animals that had been returned to food for at least 3 hrs reverted to the "well-fed" state (Fig 1C). Taken together, these results indicate that food deprivation reversibly modifies sensory integration behavior.

We then tested whether this food deprivation-evoked change in sensory integration behavior was specific to the copper repellent and diacetyl attractant used in the assay. We observed that in this sensory integration assay layout and with the diacetyl concentration used (1:2000), food-deprived animals did not cross the repellent barrier when diacetyl was paired with other repellents like fructose (with the exception of one intermediate concentration), sodium chloride, or quinine (Fig 1D). In contrast, when copper was paired with other attractants like benzaldehyde and isoamyl alcohol, food-deprived animals continued to cross the copper barrier more readily than well-fed animals (Fig 1E). Collectively, these data suggest that multiple food-associated volatile attractants can promote repellent barrier crossing in food-deprived animals. Consistently, a previous study showed that food-deprived animals are more likely to cross the repellent barrier when paired with a bacterial lawn [31]. To test whether food-deprivation differentially affected copper or diacetyl responses, we analyzed responses of these animals to varying concentrations of copper or diacetyl alone. We found that food-deprived animals crossed the copper barrier more readily than well-fed animals, suggesting that their responsiveness to copper is reduced even in the absence of an attractant (Fig 1F). In contrast, food-deprived animals did not discernably alter their attraction to diacetyl in the absence of the copper repellant, except at one intermediate concentration (Fig 1G). Given the small number of well-fed animals that cross the copper barrier alone (Fig 1F), we continued to pair copper with the diacetyl attractant for further analysis. We found that food-deprived animals were significantly more likely than well-fed animals to cross the repellent barrier above a threshold of 5 mM copper concentration (Fig 1H). To gain further confirmation of this copper-specific change, we tested food-deprived animals in a single animal copper drop assay (S2A Fig). In this assay, the response of a single animal to a drop of 1.5 mM $CuSO_4$ solution placed in its path was monitored. Most repellents can be tested in this assay with animals generating a robust avoidance response [32]. We found that food-deprived animals showed a significant deficit in their copper-avoidance response (S2B Fig). Collectively, these data show that food-deprived animals display reduced avoidance of copper repellent, which we dissected further using genetic methods and tracking software.

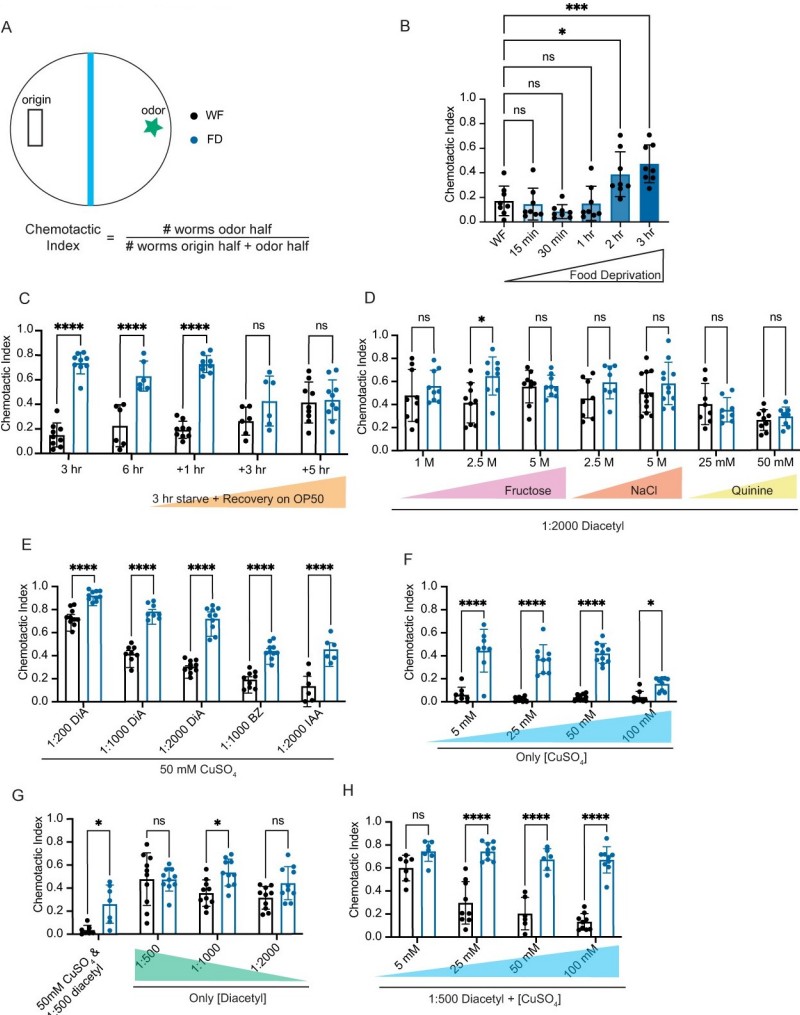

**Fig 1. Starvation reduces copper avoidance.** A) Schematic of the sensory integration assay. ~100–200 day 1 adult animals (n) are placed in the black rectangle. Blue barrier represents copper barrier (or other repellant) and star represents diacetyl or other attractant. Chemotactic Index is the number of animals that have crossed the barrier (odor side) divided by the total number of animals on the plate (odor + origin sides). Experiments with well-fed (WF) animals will appear with black dots and those with food-deprived (FD) animals will be indicated with blue dots. Unless otherwise noted, FD is 3 hrs with no food. Each dot represents a single plate (N) of animals (n). B) Animals are deprived of food for increasing periods of time (15 mins, 30 mins, 2 hrs, 3 hrs). Animals are exposed to 50 mM $CuSO_4$ repellant and 1:500 (0.2%) diacetyl attractant. N = 8. C) Sensory integration behaviors of animals that have been starved for 3 hrs and 6 hrs. Animals that have been starved for 3 hours are allowed to recover for 1, 3, or 5 hrs on OP50. Well-fed matched partners are kept on OP50 plates for the entire length of the experiment. Animals are exposed to 50 mM $CuSO_4$ repellant and 1:500 (0.2%) diacetyl attractant. N≥6. D) Animals are exposed to increasing concentrations of other repellants (Fructose, NaCl, Quinine) with the attractant 0.05% diacetyl (1:2000) in each condition N≥7. E) Animals are exposed to decreasing concentrations of diacetyl (DiA) (0.2%, 0.1% and 0.05%, or 1:200, 1:1000, and 1:2000, respectively) and other volatile attractants 0.1% Benzaldehyde (BZ) and 0.05% Isoamyl Alcohol (IAA). 50 mM $CuSO_4$ is the repellant in each condition N≥6. F) Animals are exposed to $CuSO_4$ in increasing concentrations (5 mM, 25 mM, 50 mM, 100 mM) without any attractant N≥8. G) Animals are exposed to diacetyl alone in decreasing concentrations (0.2%, 0.1%, 0.05%). Full assay (0.2% diacetyl and 50 mM $CuSO_4$) is included as a control N≥7. H) Animals are exposed to 1:500 diacetyl and increasing concentrations of $CuSO_4$ (5 mM, 25 mM, 50 mM, 100 mM) N≥6. All graphs are analyzed using a two-way ANOVA, determined to have significant differences across well-fed and food-deprived conditions. WF/FD comparisons were then performed as pairwise comparisons within each genotype or treatment as t-tests with Bonferroni corrections for multiple comparisons. * $p < 0.5$, ** $p < 0.01$, *** $p < 0.001$, **** $p < 0.0001$, ns $p > 0.05$. Error bars are S.D.

## Dynamics of risky search strategies in food-deprived animals

To analyze how food deprivation modifies animal behavior, we recorded and tracked populations of animals over 45 mins in the sensory integration assay. Individual animal trajectories were identified and used for analysis (see Methods and S3 Fig). We found that fewer well-fed animals cross the repellent copper barrier (Fig 2A) as compared to food-deprived animals (Fig 2B) during the entire 45 min sensory integration assay (example tracks in assays with copper and diacetyl together are shown in S3A–S3D Fig). To quantify this difference, we plotted cumulative net copper barrier crossings over time (Fig 2C, methods described in S4A Fig), a metric comparable to chemotactic index (S4D Fig). We found that food-deprived animals were more likely to cross the barrier at all time points (15, 30, and 45 mins) compared to well-fed animals. Thus, the differences between well-fed and food-deprived animals were not limited to specific time windows in the assay.

To further understand the dynamics of increased copper barrier crossing in food-deprived animals, we compared the probability of animal tracks being located at given distances from

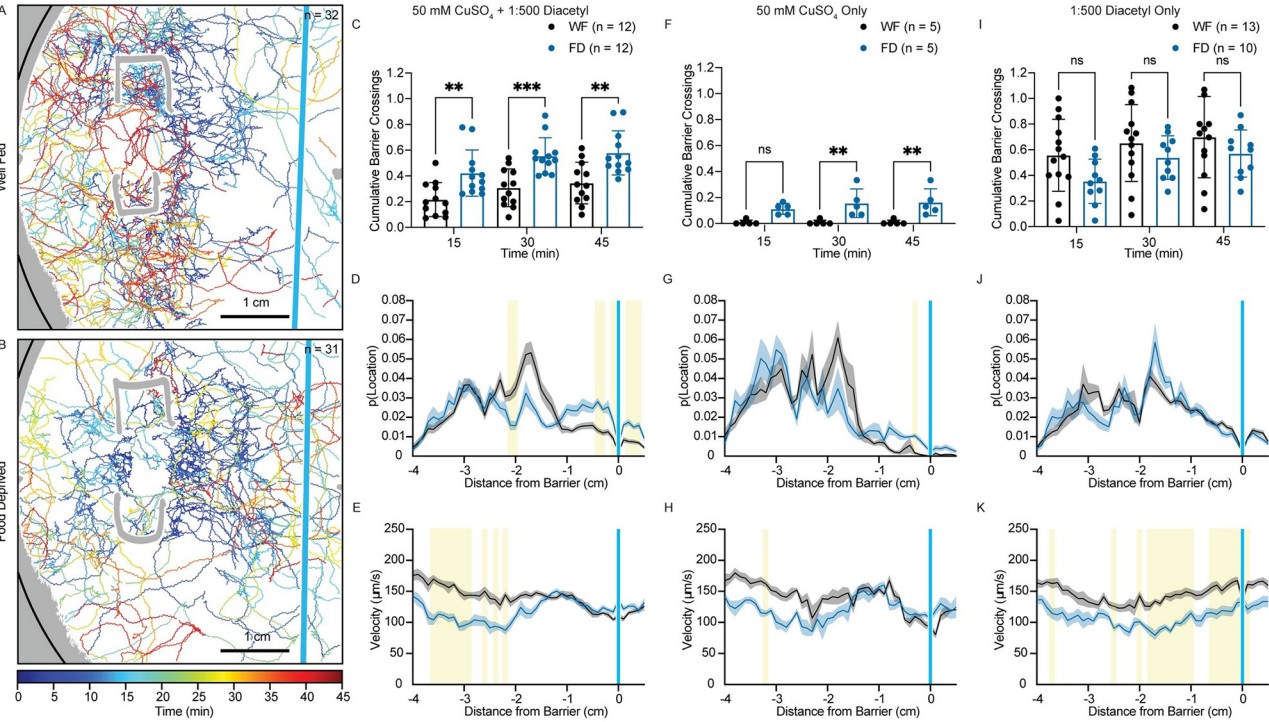

**Fig 2. Riskier search strategies in food-deprived animals.** (A) Worm tracks (n = 32) are plotted for a representative sensory integration assay of well-fed animals behaving in the presence of 50mM CuSO₄ (blue stripe) and 1 µl 0.2% diacetyl (1:500) (location out of view to the right). Regions of the plate that were not able to be tracked are in gray with the edge of the plate indicated in black. Tracks are plotted and color coded for time (0 to 45 minutes). (B) Worm tracks (n = 31) are plotted for a representative sensory integration assay of 3 hour food-deprived animals. Conditions and plotting the same as in A. (C, F, I) The fraction of mean cumulative net barrier crossings is plotted at three time points (15, 30, and 45 minutes). Well-fed (WF) animals appear with black dots and food-deprived (FD) animals are indicated with blue dots. Each dot represents a single plate of animals. C) 50 mM CuSO₄ and 0.2% diacetyl F) 50 mM CuSO₄, no diacetyl I) No copper, 0.2% (1:500) diacetyl. Graphs are analyzed using a two-way ANOVA to determine significant differences across well-fed and food-deprived conditions. WF/FD comparisons were then performed as pairwise comparisons within each time period as t-tests with Bonferroni corrections for multiple comparisons. * p<0.5, ** p<0.01, *** p<0.001, **** p<0.0001, ns p>0.05. (D, G, J) The probability of an animal being located at 1 mm binned distances from the barrier is plotted for well-fed (black) and food-deprived animals (blue). The dark line represents the mean probability of residence with the shaded areas representing the standard error of the mean. D) 50 mM CuSO₄ and 0.2% diacetyl G) 50 mM CuSO₄, no diacetyl J) No copper, 0.2% diacetyl. For each graph, multiple unpaired t-tests with Welch's correction were performed with correction for multiple comparisons with Holm-Šídák post-hoc test. Corrected p values <0.05 are indicated by yellow shading. A comprehensive list of the statistics can be found in S2 Table. (E, H, K) The mean velocity of animals as a function of distance from the barrier is plotted for well-fed (black) and food-deprived animals (blue). Conditions, plotting, and statistics are the same as in D, G, and J.

the barrier (Fig 2D, methods described in S4B Fig)). We found that food-deprived animals were nearly twice as likely to reside within +/- 0.5 cm from the copper barrier while well-fed animals were more likely to be found 2.1 to 2.2 cm from the barrier, not far from where the animals were originally placed on the assay plate (Fig 2D, statistics summarized in S2 Table). These data suggest that well-fed animals slow down or reorient upon detection of copper thereby increasing the likelihood of animals being located in regions well before the barrier. In contrast, food-deprived animals cross the barrier more frequently. To further dissect these behavioral differences, we quantified the mean velocity of worm tracks as a function of distance from the copper barrier (Fig 2E, methods described in S4C Fig). We find that well-fed animals tend to move more slowly when at distances closer to the copper barrier. Food-deprived animals are significantly slower than well-fed animals at distances far from the copper barrier (2.2–3.6 cm), but appear to accelerate to speeds matching well-fed behavior as they approach the barrier before slowing down as they reach the barrier in a manner consistent with well-fed animals (Fig 2E and S2 Table).

When this assay is run in the absence of the attractant diacetyl, we observe an increased likelihood of food-deprived animals to cross the copper barrier as compared to well-fed animals, but only to a significant extent at later time-points– 30 and 45 mins (Fig 2F). Further, the probability of food-deprived animals locomoting close to the copper barrier is higher at 0.3 cm before the barrier (Fig 2G and S2 Table) while the velocity of these animals as a function of distance to the barrier is distributed similarly to animals assayed with copper and diacetyl with the velocity of well-fed animals being significantly higher than food-deprived animals at 3.2 cm from the copper barrier (Fig 2H and S2 Table). These data suggest that food-deprived animals are more likely to cross the copper barrier than well-fed animals, even in the absence of an attractant. However, the dynamics of food-deprived animals in the absence of an attractant are different than those of animals exposed to both attractive and repellant cues. Specifically, animals are more likely to reverse and cross back over the copper barrier when copper alone is used (S4E–S4G Fig). Thus, the addition of the attractant, diacetyl, allows us to more easily interrogate this food deprivation behavior.

In the absence of copper, food-deprived and well-fed animals behaved similarly with no significant difference in the fraction of copper-less "barrier" crossings toward the diacetyl or localization on the assay plate (Fig 2I and 2J, and S2 Table). However, food-deprived animals displayed a decreased average velocity at most locations (Fig 2K and S2 Table), consistent with previous studies [33]. Collectively, these data suggest that the observed increase in food-deprived animals crossing the copper barrier is neither due to increased motility nor decreased sensitivity to volatile odorants. Rather, it may be a result of the increased tendency for food-deprived animals to pursue an unfavorable navigation strategy towards the repellent copper toxin [34] in search of food.

## Lack of food and not changes in fat drives the food-deprivation induced behavioral change

Given that the change in sensory integration behavior requires multiple hours of food deprivation, we hypothesized that metabolic signals like changes in fat content might play a crucial role. Previous studies have shown that prolonged starvation can deplete fat stores in *C. elegans*, which in turn can affect behavior [35,36]. We tested whether 3 hrs of food deprivation alters the fat content of animals. Oil-Red O (ORO), a fat-soluble dye that stains triglycerides and lipoproteins, and has been used to label and quantify fat stores in *C. elegans* (Fig 3A and 3B) [37]. We used this dye and found that 3 hrs of food-deprivation altered neither the ORO signal nor the area of the animal labeled by this stain (Fig 3C and 3D). In contrast, we observed a

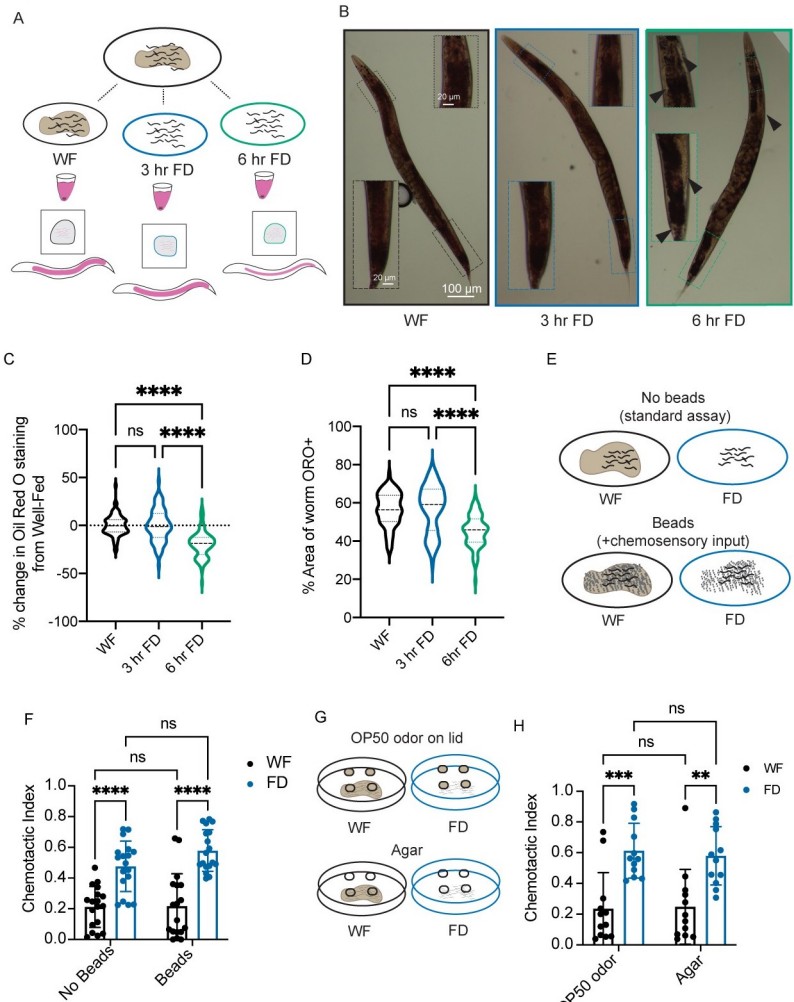

**Fig 3. Lack of food, not fat or physical interactions, drive behavioral changes.** (A) Schematic of Oil Red O experiments. Animals are raised together to day 1 of adulthood and separated into three groups: well-fed (on food), 3 hr food-deprived, and 6 hr food-deprived. Animals are stained using Oil Red O and then imaged using a color camera. (B) Representative images of well-fed (WF, black), 3 hour food-deprived (3hr FD, blue), and 6 hr food-deprived (6 hr FD, green). Inset images are shown, highlighting the regions where there is the most difference in staining. Black arrows highlight regions of no Oil Red O stain in 6 hr FD. (C) Graph showing the percent change in Oil Red O staining when compared to the average of the area of Oil Red O signal above a threshold value in the well-fed group within each independent experiment. N = 3, n>20 within each experimental treatment group. (D) Graph showing the percent of the animals' area that contains Oil Red O signal above threshold N = 3, n>20 within each experimental treatment group. Same data as in C, shown as non-normalized values. (E) A schematic representing the experiment in F, in which populations of animals are either well-fed or food-deprived in the presence or absence of Sephadex beads before performing the sensory integration assay. (F) Prior to the sensory integration assay, animals are exposed to either standard OP50 ("no beads WF") or empty plates ("no beads FD"), or Sephadex gel beads as chemosensory input. Alternatively, animals were exposed to beads and no food ("beads FD") or OP50 with Sephadex beads on top ("beads WF") for 3 hours. Animals were then exposed to standard Sensory Integration Assay set-up with 50 mM $CuSO_4$ and 1 µl of 0.2% diacetyl. N≥18. (G) A schematic representing the experiment in H, in which populations of animals are either well-fed or food-deprived in the presence of OP50-containing agar plugs on the lid of the plate or agar alone plugs on the lid of the plate before performing the sensory integration assay. (H) Prior to the sensory integration assay, animals are exposed to either standard OP50 empty plates, covered with lids containing either agar plugs (agar) or agar plugs with OP50 lawns (OP50 odor) as a chemosensory input for 3 hours. Animals were then exposed to standard Sensory Integration Assay set-up with 50 mM $CuSO_4$ and 1 µl of 0.2% diacetyl. N = 12 per condition. C and D were analyzed using Welch's ANOVA test with Dunnett's multiple comparisons test. * p<0.5, ** p<0.01, *** p<0.001, **** p<0.0001, ns p>0.05. F and G were analyzed using a full model two-way ANOVA, determined to have significant differences across well-fed and food-deprived conditions but no difference between "bead"/"no bead" groups or "odor"/"agar" groups. Those comparisons are shown to indicate no difference between "beads" and "no beads". Pairwise comparisons within each treatment were performed as t-tests with Tukey's multiple comparisons test. Error bars are S.D.

significant change in the both the intensity of the signal and area of animal stained in 6-hr food-deprived animals, consistent with previous studies [38]. These data suggest that changes in sensory integration behavior, which occur after 3 hours of food deprivation, are likely to be independent of fat metabolism as measured by Oil-Red O staining.

Next, we sought to identify the relevant aspects of the bacterial experience contributing to the food deprivation-triggered behavioral change. *C. elegans* has been shown to evaluate multiple aspects of the food experience, including changes in food distribution, oxygen and carbon dioxide concentrations, small molecule metabolites, and others [39–41]. To uncouple the tactile and chemosensory input of the bacteria from the nutritional value of ingesting bacteria, we analyzed the effect of using Sephadex gel beads on animal behavior. Animals exposed to gel beads experienced the tactile input, but not the nutritional value of food (Fig 3E) [15]. Notably, we found that animals exposed to Sephadex beads in the absence of *E. coli* (OP50) for 3 hrs behaved similarly to food-deprived animals in the sensory integration assay (Fig 3F) suggesting that mechanosensory input is not involved in this food-deprivation invoked behavior. Next, we tested whether chemosensory information from bacteria can affect animal behavior in the sensory integration assay. We exposed animals to the volatile odors of OP50 using agar plugs on the lid of the dish prior to performing the assay (Fig 3G). Animals exposed to the volatile odors of OP50 in the absence of food for 3 hrs behaved similarly to food-deprived animals exposed to agar plugs with no OP50 volatile odors (Fig 3H). Together, these results suggest that the absence of volatile chemosensory and mechanosensory cues do not reduce the animal's copper avoidance behavior. Rather, the lack of food in the *C. elegans* intestine may be causing the observed food-deprivation behavior.

## Transcription factors mediate food deprivation-induced behavioral change

Our study suggests that the lack of food inside the animal is responsible for the reversible reduction in copper avoidance. To gain insights into the underlying molecular machinery, we investigated the role of nutritional-responsive transcription factors in the sensory integration assay. In mammalian cells, glucose is rapidly converted to glucose-6-phosphate, whose levels are sensed by two basic-helix-loop-helix-leucine zipper transcription factors, MondoA and ChREBP (Carbohydrate Response Element Binding Protein). In well-fed conditions, MondoA binds excess glucose-6-phosphate and Mlx (Max-like protein X) and translocates to the nucleus where it activates transcription of glucose-responsive genes. In the absence of glucose, MondoA remains in the cytoplasm [42,43] (Fig 4A). *C. elegans* orthologs of MondoA and Mlx have been identified as MML-1 and MXL-2, respectively [44]. MML-1/MondoA has previously been shown to translocate into the intestinal nuclei under well-fed conditions (Fig 4A) [45]. We predicted that *mml-1* mutants would be unable to sense the lack of food and thereby unable to reduce copper sensitivity after food deprivation. Consistently, we found that *mml-1*, but not *mxl-2* mutants were defective in their integration responses after food deprivation (Fig 4B). We then tested whether food deprivation alters the sub-cellular localization of the MML-1 protein. We monitored the GFP fluorescence under well-fed and food-deprived conditions in a *mml-1* knockout transgenic animal expressing GFP fused to the full-length coding sequence of MML-1/MondoA under well-fed and food-deprived conditions [45]. We found that 3 hrs of food-deprivation resulted in an increased translocation of MML-1/MondoA from the nucleus to the cytoplasm of the intestinal cells (Fig 4C and 4D). We suggest that this cytosolic MML-1/MondoA reduces copper avoidance by modifying signaling between tissues.

Previous studies have shown that MML-1 regulates the activity and nuclear localization of a second bHLH transcription factor HLH-30 (*C. elegans* TFEB, Fig 4A) [46]. In multiple animal models, HLH-30/TFEB functions as a key regulator of longevity pathways by promoting

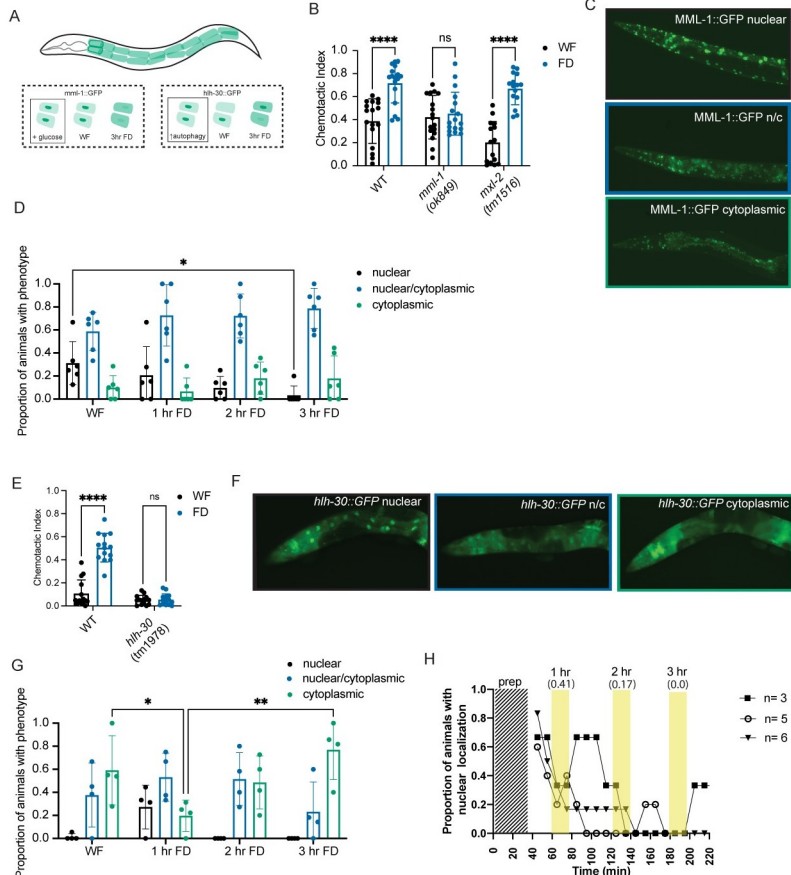

**Fig 4. _mml-1_ and _hlh-30_ are required for sensory integration change upon food deprivation, correlated with shifts in their intestinal localization.** (A) Schematic showing the 20 intestinal cells in a day 1 adult _C. elegans_. Our findings for _mml-1_::_gfp_ and _hlh-30_::_gfp_ transgenic animals are shown in the dotted box, while previously published paradigms are within the solid line box. Addition of glucose has been shown to induce nuclear localization of MondoA. Autophagy has been shown to increase nuclear localization of HLH-30. (B) Standard sensory integration assay with _mml-1(ok849)_ and _mxl-2(tm1516)_ and wildtype controls. N = 20. (C) Representative images of MML-1::GFP localization in day 1 adult animals (data quantified in D). All images were collected with the same exposure time and laser power. (D) Intestinal MML-1::GFP expression in animals during static timepoints food deprivation. Only intestinal expression was characterized as "nuclear", "nuclear/cytoplasmic", or "cytoplasmic". Each dot represents the proportion of animals within an experiment with the phenotype. N = 6, n = 296.(E) Standard sensory integration assay with _hlh-30(tm1978)_ mutant animals and wildtype controls. N = 9. (F) Representative images of HLH-30::GFP localization in day 1 adult animals (data quantified in G). All images were collected with the same exposure time and laser power. (G) Intestinal HLH-30::GFP expression in animals during static timepoints of food deprivation. Only intestinal expression was characterized as "nuclear", "nuclear/cytoplasmic", or "cytoplasmic". Each dot represents the proportion of animals within an experiment with the phenotype. N = 3, n = 149. (H) Intestinal HLH-30::GFP expression in animals during time lapse imaging. The proportion of animals (n = 3, n = 5, n = 6) with nuclear localization are plotted over time, with images taken every 10 minutes. The areas shaded in yellow correspond to the timepoints that match those in the separate experiments in Fig 4G, with the average of the timepoints within that period of time in parentheses. The shaded region labeled "prep" denotes time that the animals are off food but cannot be imaged due to preparation constraints. B and E were analyzed using two-way ANOVA, determined to have significant differences across well-fed and food-deprived conditions. WF/FD comparisons were then performed as pairwise comparisons within each genotype or treatment as t-tests with Bonferroni's multiple comparisons test. D and G were analyzed using Two-Way ANOVA, determined to have significant differences across localization and an interaction between time of food deprivation and localization. Within each localization group, pairwise comparisons were performed across each time point and tested for significance using Tukey's multiple comparisons test. * $p<0.5$, ** $p<0.01$, *** $p<0.001$, **** $p<0.0001$, ns $p>0.05$. Error bars are S.D.

autophagy and lysosome biogenesis [47–50]. We tested whether HLH-30/TFEB was also required for food deprivation-evoked change in sensory integration. We found that, unlike wild-type animals, *hlh-30* null mutants did not show a change in their behavior after food-deprivation in the sensory integration assay (Fig 4E), but were mobile in the absence of a copper barrier (S1 Table). We then tested whether the subcellular localization of HLH-30/TFEB was also affected by food deprivation. We observed an initial decrease in cytosolic GFP fluorescence at 1 hour of food-deprivation, with a concomitant increase in nuclear localization in HLH-30::GFP transgenic animals [47] (Fig 4F and 4G). Subsequently, at 3 hrs of food-deprivation, we found a robust increase in cytosolic HLH-30::GFP fluorescence with a decrease in nuclear localization at 2 and 3 hrs of food deprivation. Further analysis of HLH-30::GFP animals throughout an *in vivo* time course of food deprivation suggested that HLH-30 nuclear intestinal localization was dynamic (Fig 4H and S3 Movie), implying a complex role for this transcription factor. Moreover, the change in localization of HLH-30 during food deprivation corresponded to the timing of behavioral changes (after 1 hour, Fig 1B), suggesting that HLH-30 might transcribe a "hunger" signal. Collectively, these data show that both MML-1 and HLH-30 change their localization in response to food-deprivation and are required for behavioral change in sensory integration.

## Intestine-to-neuron signaling involves insulin signaling

Previous studies have shown that the *C. elegans* intestine is a major site for the transcriptional regulation of insulin-like peptide genes in response to starvation [51]. In addition, HLH-30/TFEB has been shown to act upstream of the insulin-signaling pathway in regulating the expression of neuronal chemoreceptor genes [52]. The *C. elegans* genome encodes about 40 insulin-like peptides [53] and all of these ligands are thought to bind and signal via a single tyrosine kinase DAF-2 receptor [54,55]. We hypothesized that insulin-like peptides might also act downstream of HLH-30/TFEB in relaying food status signals from the intestine to other tissues. Consistent with our hypothesis, multiple insulin-like peptides including INS-3, INS-4, INS-6, INS-10, INS-11, INS-17, INS-18, INS-23, and INS-31 contain HLH-30/TFEB binding sites in their promoters [52]. In addition, INS-7, INS-8, and INS-37 have been shown to affect the subcellular localization of HLH-30/TFEB in the *C. elegans* intestine after mating [56] (summarized in Fig 5A). We tested mutants in these insulin-like peptide genes for their ability to alter sensory integration behavior after food deprivation. Some alleles of *ins-3* (*ok2488* and *tm3603*), *ins-4* (*ok3534*), and *ins-18* (*tm339*) show altered chemotactic behaviors, where well-fed and food-deprived indices are similar (Fig 5B). However, upon testing additional alleles we find that animals carrying *ins-3(ok2487)*, *ins-4(tm3620)*, or *ins-18(ok1672)* alleles as well as *ins-6(tm2416)*, *ins-10(tm3498)*, *ins-11(tn1053)*, *ins-17(tm790)*, *and ins-23(tm1875)* displayed significantly different chemotactic indices when food deprived, making their responses similar to wild-type animals (Fig 5B). In contrast, we found that null mutants in the insulin-like peptide *ins-31*(*tm3543*) were unable to respond to food deprivation. Specifically, these mutant animals did not display an increased ability to cross the repellent copper barrier when food deprived (Fig 5B), implying that INS-31 might be a candidate signal relaying food status signals. Also, mutants in other insulin-like peptides that affect the localization of HLH-30/TFEB, *ins-7(tm2001)*, *ins-8(tm4144)*, *and ins-37(tm6061)*, were similar to wild-type animals in their ability to cross the copper barrier in both well-fed and food-deprived conditions (Fig 5C). To directly assess whether INS-31 is produced or secreted from the intestinal cells, we generated a rescue construct to drive expression of *ins-31* cDNA in either the intestine (from the *gly-19* promoter) or in neurons (from the *H20* promoter) in the background of *ins-31(tm3543)* mutant animals. We observe that intestinal, but not neuronal-selective expression of *ins-31*

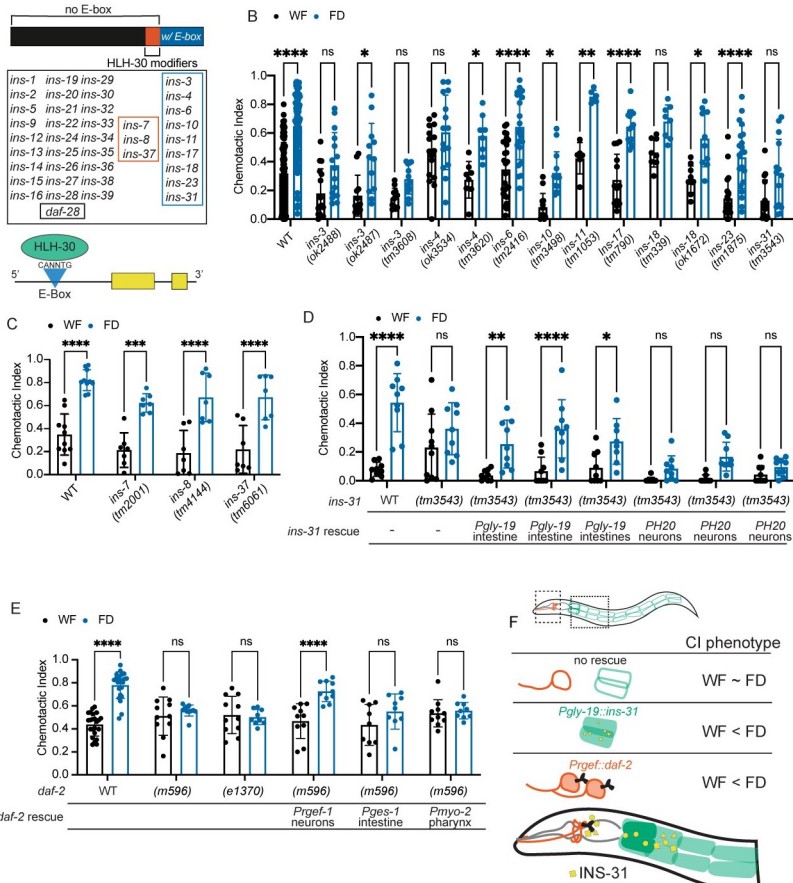

**Fig 5. Sensory integration changes require HLH-30-regulated insulin while *daf-2* is required in neurons.** (A) HLH-30 interacts with *C. elegans* insulin peptides. Of the 40 insulin-like peptides encoded in the *C. elegans* genome, 22% have an HLH-30 binding motif (CANNTG E-box, blue) in the 5' UTR (< 300bp upstream of start site) [52]. 7% of insulins have been shown to regulate the localization of HLH-30 but do not contain an E-box (orange, "HLH-30 modifiers"). An illustration of a representative insulin peptide with two yellow exons and an upstream E-box with HLH-30 initiating transcription. (B) All insulins known to contain an HLH-30 binding motif in the 5' UTR were tested using the standard sensory integration assay. When available, more than one allele was tested (N≥8) for each insulin, with wildtype (N2) animals tested with each mutant. (C) Insulins previously shown to regulate HLH-30 localization (*ins-7*, *ins-8*, *ins-37*) were tested using the standard sensory integration assay alongside wildtype (N2) control. N≥7. (D) *ins-31* mutants and tissue-specific rescues are tested in the standard sensory integration assay. N ≥8 for each strain tested alongside wildtype N2. *ins-31* is rescued in neurons and intestines using tissue-specific promoters. (E) *daf-2* mutants and tissue-specific rescues are tested in the standard sensory integration assay N ≥ 9 for each strain tested alongside wildtype N2. *daf-2* is rescued in neurons, intestines, and pharynx using tissue-specific promoters. (F) Schematic showing requirement of *ins-31* in the intestine and *daf-2* in neurons. CI phenotype means Chemotactic Index phenotype, where wildtype animals display a chemotactic index of WF < FD. All graphs were analyzed using a two-way ANOVA, determined to have significant differences across well-fed and food-deprived conditions. WF/FD comparisons were then performed as pairwise comparisons within each genotype or treatment as t-tests with Bonferroni's multiple comparisons test. * $p<0.5$, ** $p<0.01$, *** $p<0.001$, **** $p<0.0001$, ns $p>0.05$.

cDNA was sufficient to restore wild-type behavior to the *ins-31(tm3543)* null mutants. Also, neuron-selective expression of *ins-31* cDNA reduces chemotactic behaviors in both well-fed and food-deprived animals (Fig 5D), indicating that aberrant neuronal expression of this transgene might affect chemotaxis behaviors. Taken together, these data suggest that the *C. elegans* intestine likely releases INS-31 to relay hunger information to other tissues.

Next, we probed the role of the insulin receptor, DAF-2, in affecting 3 hour-food deprivation evoked changes in sensory integration. Consistent with our analysis of mutants in various

insulin-like peptide genes, we found that two different alleles (*m596* and *e1370*) in the insulin receptor, DAF-2, were also defective in their response to food deprivation (Fig 5E). To localize the site of DAF-2 action, we analyzed the effect of rescuing this receptor in different tissues. We found that re-expressing *daf-2* under neuronal (*rgef-1*), but not intestinal (*ges-1*) or pharyngeal muscle (*myo-2*) promoters [57] restored normal behavior to *daf-2(m596)* mutants (Fig 5E). Taken together, these results suggest that neuronally expressed DAF-2 receptors might detect INS-31 peptides released from the intestine (Fig 5F).

## DAF-2 signaling pathway components affect food-deprived animal behavior

We next sought to identify components of the DAF-2 signaling pathway (Fig 6A) that were required to alter food-deprivation evoked change in sensory integration. We observed that mutants in the insulin-signaling pathway components including the lipid phosphatase (*daf-18*, PTEN suppressor), 3-phosphoinositide-dependent kinase 1 (*pdk-1*), serine/threonine kinases AKT-1, AKT-2 (*akt-1*, *akt-2*), and the FOXO family transcription factor *daf-16* performed normally in the sensory integration assay after food deprivation (Fig 6B) [58,59]. In contrast, mutants in the phosphoinositide 3-kinase (PI3K) *age-1* were defective in their copper responsiveness after food deprivation (Fig 6B), yet still mobile in the assay in the absence of a copper repellant (S1 Table). Collectively, we suggest that food deprivation likely engages DAF-2 and

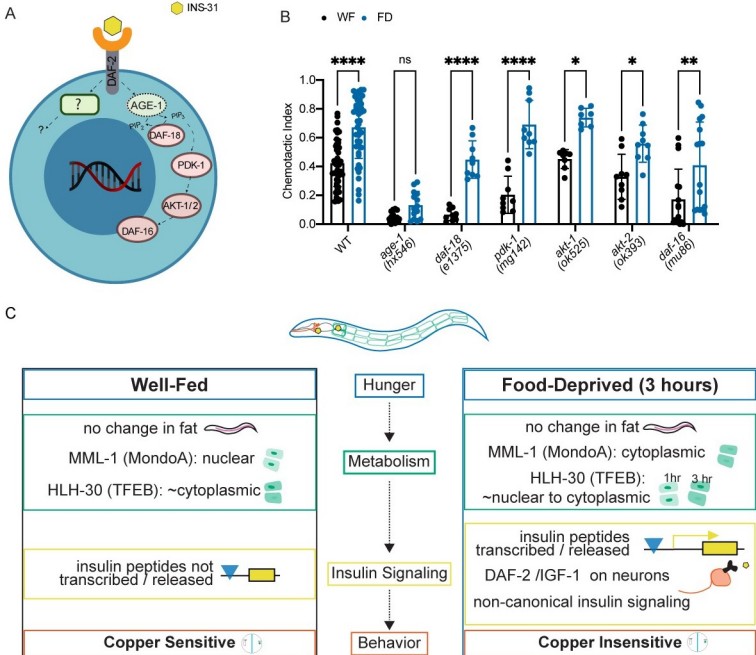

**Fig 6. Insulin-signaling pathway acting downstream of DAF-2 receptors.** (A) Schematic of a neuron's *daf-2*-mediated canonical and non-canonical insulin signaling. Summary of the findings in B. Created with Biorender.com (B) Standard sensory integration assay performed with mutants in the canonical insulin signaling pathway (*age-1*, *daf-18*, *pdk-1*, *akt-1*, *akt-2*, and *daf-16*,), alongside wildtype N2 N≥7. (C) Summary of data and proposed model through which food deprivation alters animal behavior. B was analyzed using two-way ANOVA, determined to have significant differences across well-fed and food-deprived conditions. WF/FD comparisons were then performed as pairwise comparisons within each genotype or treatment as t-tests with Bonferroni's multiple comparisons test. * p<0.5, ** p<0.01, *** p<0.001, **** p<0.0001, ns p>0.05.

PI 3-kinase signaling, likely in neurons, to alter the animal's copper responsiveness allowing it to cross the copper barrier more readily.

## Discussion

We used *C. elegans* as a model to understand how food deprivation modifies behavior. We show that food-deprived animals reversibly alter their behavior by reducing their repellent responsiveness, allowing them to traverse potentially toxic environments in their search for food. The *C. elegans* intestine senses the lack of food leading to cytosolic MML-1 and nuclear HLH-30, which in turn might promote the release of the insulin-like peptide INS-31. These intestine-released peptides likely bind neuronal DAF-2 receptors and their downstream PI 3-Kinase to reduce copper avoidance and alter behavior (Fig 6C).

Multicellular animals sense and regulate glucose homeostasis at multiple levels. While insulin and glucagon maintain constant levels of circulating glucose, the Myc-family transcription factors are used within cells. Glucose uses cell membrane-localized transporters to enter cells, where it is rapidly converted into glucose-6-phosphate [60]. This intermediate metabolite is sensed by the Myc-Max complex, which binds glucose-6-phosphate and translocates to the nucleus where it regulates the transcription of glucose-responsive genes [42]. While the role of ChREBP/MondoA-Mlx-glucose-6-phosphate complex in regulating transcription is well studied [43,61,62], the role of these proteins in the cytoplasm remains poorly understood. We show a specific role for MML-1 (MondoA homolog), but not MXL-2 (Mlx homolog) in reducing copper avoidance after food deprivation. Additionally, we show that HLH-30, an ortholog of TFEB, is also required for attenuating copper responsiveness after food deprivation. Intriguingly, MML-1/MondoA and HLH-30/TFEB are both basic helix-loop-helix transcription factors and have been shown to act in concert to modify signaling networks and affect global states like reproduction or survival [46,63]. In *C. elegans* and mammals, HLH-30/TFEB accumulates in the nucleus during nutrient deprivation to induce autophagy and lysosomal activity [64]. Our findings indicate that HLH-30/TFEB's activity is highly dynamic, with altered nuclear localization between 45–60 mins post-food deprivation. Also, longer periods of food deprivation or alternative methods of acute stress likely result in differential HLH-30/TFEB engagement and localization patterns. We suggest that MML-1/MondoA accumulation in the cytoplasm and translocation dynamics of HLH-30/TFEB (in food-deprived animals) might enable the intestine to release peptide(s) relaying a "lack of glucose" signal to other tissues.

Helix-loop-helix transcription factors in *C. elegans* like MML-1/MondoA and HLH-30/TEFB are known to bind similar E-box elements (CACGTG) and have large overlap in their target gene expression [44,46]. Additionally, previous studies have identified multiple insulin-like peptide genes whose expression is regulated by HLH-30/TFEB and other insulin-like peptide genes, which can affect the subcellular localization of HLH-30/TFEB [52,56]. We screened this subset of insulin-like peptide genes to identify a candidate that might relay food status signals from the intestine to other tissues. The *C. elegans* intestine has been previously shown to be a key tissue where the transcription of insulin peptide genes is regulated [51,55,65]. From these data, it is possible for insulin-like peptides to play a role in the behavioral shift observed in food-deprived animals in a manner independent of MML-1/MondoA and/or HLH-30/TEFB. However, we have directly demonstrated that our candidate insulin-like peptide, INS-31, is required in the intestine to affect food-deprivation evoked behavioral change. Interestingly, *ins-31* has also been shown to be involved in maintenance of specific gut microbiota signatures, indicating that *ins-31* is active in determining intestinal states [65].

We show that neurons use the tyrosine kinase insulin receptor (DAF-2) to integrate these signals. Based on our current model, we are unable to confirm whether INS-31 is acting as an

agonist or antagonist of DAF-2; evidence from other groups suggests that it might play multiple roles in engaging with the insulin receptor [55]. We also define the role of additional insulin signaling pathway components in the sensory integration behavior. While AKT kinase -1 and -2, PDK-1, PTEN (DAF-18) and FOXO (DAF-16) are not required, we show that AGE-1 (PI-3 Kinase) may be required downstream of the DAF-2 receptor to affect copper avoidance after food deprivation. HLH-30/TFEB has been shown to engage with DAF-16 to promote stress responses [66]. However, since *daf-16* mutant animals behave similarly to wild-type animals in their sensory integration behaviors after food-deprivation (Fig 6B), our data suggest that HLH-30 might be acting independently of DAF-16 to modify copper avoidance.

Multiple studies have also highlighted the role of insulin signaling in relaying starvation-related signals to various neurons. Starvation has been shown to decrease the secretion of INS-18 from the intestine, which antagonizes DAF-2 receptor in ADL neurons and modifies pheromone-mediated behaviors [20]. Starvation is also associated with increased octopamine signaling, which transforms $CO_2$ attraction to repulsion in starved animals [18]. Additionally, starvation has been shown to recruit ASG neurons to cooperate with ASE neurons and drive avoidance to high salt [19]. Collectively, we speculate that this intestine-to-neuron insulin signaling pathway likely alters ASI, ASH, or ADL chemosensory neuronal activity and altered copper sensitivity. In support of this hypothesis, starvation has been shown to increase ASI neural activity in response to food-stimuli [67] and increased ASI activity suppress ASH neural responses [68]. These studies are also consistent with previous studies showing that ASI neurons playing a crucial role in modifying behavior after 6 hours of food deprivation [17,69]. Taken together, we speculate that food deprivation leads to an increase in insulin signaling from the intestine to neurons, which alters neuronal activity and reduces the animal's sensitivity to copper, allowing it to cross the barrier more readily. More broadly, these studies link transcription factors and insulin signaling from the intestine to neurons to modify sensory behavior, a mechanism likely conserved across species.

## Methods

### Strains

*C. elegans* strains were grown and maintained under standard conditions [70]. All strains used are listed in S1 Table.

### Behavior assays

All animals were grown to adulthood on regular nematode growth medium (NGM) plates seeded with OP50 ($OD_{600}$ ~ 0.2) before they were washed and transferred to new food (standard NGM plates seeded with OP50) or food-free plates (standard NGM plates) respectively for the indicated duration. Sephadex beads (G-200) were added to both the empty NGM plate and the OP50 lawn in experiments for Fig 3E and 3F. Sensory integration assays were performed on 2% agar plates containing 5 mM potassium phosphate (pH 6), 1 mM $CaCl_2$ and 1 mM $MgSO_4$, made the day before the experiment. Repellent gradients (including $CuSO_4$ (Copper (II) sulfate pentahydrate, Sigma 209198), NaCl, fructose (D-(-) Fructose Sigma F0127), and quinine (Sigma 22620)) were established by dripping 25 μl of solution, dissolved in water, across the midline of the plate [30]. This solution was allowed to dry overnight: copper, NaCl and quinine barriers were allowed to dry overnight, while fructose was applied 3-5 minutes before the assay began. Before the assay, 1 μl of 1M sodium azide in water (Sigma 71289) was placed on the opposite side of the chemotaxis plate. Prior to the assay, the animals were washed from the food or food-free plates into Eppendorf tubes. Each treatment group was serially washed once with M9+$MgSO_4$ and 3 times with Chemotaxis buffer (5 mM potassium

phosphate (pH 6, Fisher BP362 monobasic and Fisher BP363, dibasic), 1 mM $CaCl_2$ (Sigma C1016), and 1 mM $MgSO_4$ (Sigma M7506)) before being transferred to the assay plates. Glass Pasteur pipets were used to prevent loss of animals sticking in plastic pipette tips. Immediately after plating 100–200 animals in a small drop of chemotaxis buffer, 1 μl of attractant was placed on the opposite side of the chemotaxis plate. Attractants used were diacetyl (2,3-Butanedione Sigma 11038), Isoamyl alcohol (3-methyl-1-butanol, Sigma 77664), and Benzaldehyde (Sigma 418099) diluted in Ethanol. The small drop of animals was dabbed gently with the edge of a Kim wipe and the lid was immediately replaced. After 45 minutes or at indicated times, the integration index was computed as the number of animals in the odor half of the plate divided by the total number of animals on the plate. For each experiment, at least two plates were tested each day with experiments performed on at least three different days. Unless otherwise noted, the repellant is a dried stripe of 25 μl 50 mM $CuSO_4$ (Copper (II) sulfate pentahydrate, Sigma 209198) in water and the attractant is 1 μl 0.2% diacetyl (2,3-Butanedione Sigma 11038) (diluted in 100% ethanol).

## Statistics

For sensory integration, experiments were performed at least 3 times with at least 2 plates per genotype/condition (unless otherwise noted). For strains with extrachromosomal arrays, only animals expressing the co-injection markers were counted. Every condition was performed with N2 (wildtype) controls at the same time. Unless otherwise noted two-way ANOVAs with post-hoc Bonferroni-corrected multiple comparisons were performed across WF/FD conditions, only if the factor was significant. For all figures, p values are represented by: * $p<0.05$, ** $p<0.01$, *** $p<0.001$, **** $p<0.0001$. Statistics were performed in GraphPad Prism 9.

## Single animal avoidance assay: Copper drop test

Experiments were performed as previously described [32]. Briefly, animals are moved from an OP50 plate to a food-free assay plate. A capillary tube is used to deliver a drop of test compound (1.5 mM $CuSo_4$) 0.5–1 mm away from the head of the animal and its responses scored. Positive avoidance indicates an animal executing a large reversal and omega bend within 3 secs of sensing the test compound. Five animals are tested per condition with each animal exposed to 10 drops and the percent avoidance is plotted. Assay is replicated at least three times by an investigator who is blind to the conditions being tested.

## Tracking

Sensory integration behavior assays using 50 mM $CuSO_4$ (Copper (II) sulfate pentahydrate, Sigma 209198) in water and 1 μl 0.2% diacetyl (2,3-Butanedione Sigma 11038) attractant (1:500 in 100% Ethanol) were performed with well-fed and food-deprived animals. Animal behavior was recorded for 45 minutes using a Pixelink camera (1024x1024 pixels at 3 frames per second). The imaging field of view was approximately 47 mm x 47 mm. WormLab software (MBF Bioscience) was used to identify and track the midpoints of animals in each video. Custom MATLAB software (https://github.com/shreklab/matty-et-al-2022) was used to further clean the data (i.e. remove putative tracks that did not correspond to animal behavior) and analyze individual tracks. Tracks were excluded if they met any of the following criteria: 1) overlapped with shadows or markings; 2) lasted less than 10 seconds; 3) travelled fewer than 30 pixels$^2$; or 4) traveled less than 10 pixels in any direction. Valid animal tracks were then plotted (Fig 2A and 2B, and S3A–S3D Fig) and analyzed as described below.

## Tracking analysis

The number of animals in each experiment is estimated from the maximum number of simultaneous tracks identified in a single frame. An average of 27.2 ± 13.9 animals were assayed across all conditions. Because the field of view does not encompass the entire plate, the number of tracks identified in each frame decreases over time as animals crawl to other regions of the plate. Further, a chemotactic index could not be computed as animals that crossed the copper barrier could not be tracked at later timepoints. Therefore, in order to quantify the number of animals crossing the copper barrier as a function of time, the number of unique tracks terminating at either side of the marker-demarcated copper barrier were A) identified, B) normalized by the estimated number of animals in each experiment, and C) categorized into two groups: 1) tracks moving in the forward direction (left to right) towards the direction of the odor; and 2) tracks moving in the reverse direction (right to left) away from the odor. Tracks that both started and ended at either side of the copper barrier–indicating a reversal–were discounted. To calculate the number of net barrier crossings in the direction of the odor, the number of reverse crossings were subtracted from the number of forward crossings (S4A Fig). These three metrics–forward, reverse, and net barrier crossings–were calculated cumulatively for all conditions at three different time points—15, 30, and 45 mins (Fig 2C, 2F and 2I; and S4E–S4G Fig). To verify that the net cumulative barrier crossings was comparable to the chemotactic index, we assayed well-fed and food-deprived animals in the sensory integration assay and found that animal behavior was comparable using either metric (Fig 2C and S4D Fig). To better understand animals' avoidance of copper and attraction to diacetyl, the probability of an animal residing at a particular distance from the barrier was calculated for 1 mm bins. The total number of tracked midpoints at each time point located in each 1 mm bin was summed and divided by the total number of tracked midpoints across all bins (Fig 2D, 2G and 2J; and S4B Fig). Additionally, animal velocity was calculated by computing the Euclidean distance of a track over a 2 sec window. In each video, the average velocity of all tracks was computed as a function of distance from the copper barrier in 1 mm bins (Fig 2E, 2H and 2K; and S4C Fig).

## Visualizing copper gradients

Copper sulfate gradients were visualized using 1-(2-Pyridylazo)-2-naphthol (PAN, Sigma 101036). Plates with 25 μl of 5 mM, 25 mM, 50 mM and 100 mM $CuSO_4$ dripped down the midline were dried overnight. 1 ml of 0.01% PAN indicator was added to plates the next day and allowed to dry. The plates with PAN indicator were incubated overnight and imaged the following day to allow for saturation of the signal. Images and quantification of the copper barrier is shown in S1 Fig.

## Fat quantification

Oil red O staining was conducted as previously described [37]. Briefly, 10–20 N2 adults were allowed to lay eggs for 1 hr on NGM plates seeded with OP50. The adults were removed and eggs were allowed to develop for 3 days. These day-1 adult animals were either removed from food and placed on an empty NGM plate for 3 hrs or 6 hrs or placed on a new plate with OP50 food. 5 mg/mL Oil Red O (Sigma, O9755) in 100% isopropanol was prepared as a working solution and diluted 3:2 in 60% isopropanol on the day before use. Mixture was kept from the light and filtered using a 0.2 μm cellulose acetate syringe filter and allowed to mix on a rocker overnight. Animals were washed off plates with PBST (PBS + 0.01% Triton X-100 (Sigma, X100) at the appropriate times and washed once. Animals were fixed in 40% isopropanol and shaken at room temperature for 3 mins. Isopropanol was removed and 600 μl of the Oil Red O

diluted solution was added to each tube. Each tube was nutated for 2 hrs at room temperature, away from light. Animals were washed once with PBST and nutated for another 30 mins. Animals were washed once more and prepared for imaging. Approximately 20 animals from each treatment group were pipetted onto a microscope slide and covered with a coverslip. Images were collected on upright Zeiss Axio Imager.M2 at 10X using an AxioCam 506 Color camera. Images were quantified using color deconvolution in ImageJ, normalized to background and an unstained region of an animal. Within each experiment, the same thresholds were used across treatments. Approximately 20 animals were quantified within each condition on each experimental day, performed across three different days.

## Imaging

Transgenic animals (*hlh-30*::*gfp* and *mml-1*::*gfp*) were grown to day 1 adulthood (3 days post hatching) via a one-hour hatch off on standard NGM plates seeded with OP50. Animals were picked onto empty NGM plates for 1, 2, and 3 hrs for food deprivation or placed on a new NGM plates with OP50. Animals were picked onto thin agar pads on microscope slides and anesthetized with 100 μM tetramisole hydrochloride (Sigma-Aldrich L9756) immediately prior to imaging. Animals were imaged at 10X using an upright Zeiss Axio Imager M2. At least 12 animals per group on three different days were imaged and qualitatively analyzed for localization to primarily cytoplasmic, nuclear, or both in intestinal cells, with the investigator blind to food deprivation status.

## Time course imaging and analysis

Transgenic animals (*hlh-30*::*gfp*) were grown to day 1 adulthood (3 days post hatching) via a one-hour hatch off on standard NGM plates seeded with OP50. Animals were picked onto a 96 well plate (Corning 3603) with 25 μl of low melt agarose in each well. Animals were anesthetized with 10 μM tetramisole hydrochloride and imaged on a Zeiss CSU Spinning Disk Confocal Microscope with a 10X objective for 2.5 hrs, collecting images every 10 mins using 5 μm stacks. Images were converted to maximum intensity projections and scored for nuclear localization using the same metrics as the static timepoints. The proportion of animals with fully nuclear localization of HLH-30::GFP are counted at each time point and displayed in Fig 4H. Due to the constraints of the imaging set-up, the first 40 mins of food deprivation are lost. A sample video is provided in S3 Movie.

## Molecular biology and transgenics

Plasmids were ordered from Epoch Biosciences to generate IV958-IV963. Tissue specific expression was achieved with the promoter *H20* for neurons and promoter *gly-19* for the intestine [71,72]. For all experiments, a splice leader (SL2) fused to *mcherry* transgene was used to confirm expression of the gene of interest in either specific cells or tissues. Germline transformations were performed by microinjection of plasmids [73] at concentrations between 50 and 100 ng/μl with 2.5 ng/μl of *myo-2*::*mcherry* as a co-injection marker.

## Supporting information

**S1 Fig (to accompany Fig 1). Spread of copper sulfate CuSO₄ on agar plates visualized using 1-(2-pyridylazo)-2- naphthol.** (A-E) 25 μl of (A) water as control, (B) 5 mM CuSO₄, (C) 25 mM CuSO₄, (D) 50 mM CuSO₄, and (E) 100 mM CuSO4 was dripped and dried overnight along the midline of the plate to form a copper gradient. PAN indicator (1-(2-pyridylazo)-2-naphthol) distributed over the entire plate shows a gradient of orange-red upon

chelation with copper ions. (F) Measured width of colored area with each data point representing the average width, error bars indicate SEM. n = 9.
(TIF)

**S2 Fig (to accompany Fig 1). Food-deprived animals fail to avoid copper in single animal drop test.** (A) Schematic for dry drop test shown in B. ~300 nL of 1.5 mM CuSO4 is dropped ~1 mm away from the animal's forward motion. Turning away or backing up is considered "avoidance" and given a score of 1. Heading toward the dried drop is considered "no avoidance" and given a score of 0. (B) Quantification of the dry drop test. Food-deprived (FD) animals were starved for 3 hours. Each dot represents the average of ten trials (drops) for a single animal, n = 15. Analyzed with an unpaired t-test $^*$ p<0.5, $^{**}$ p<0.01, $^{***}$ p<0.001, $^{****}$ p<0.0001, ns p>0.05. Error bars are S.D. Created with Biorender.com.
(TIF)

**S3 Fig (to accompany Fig 2). Example traces for copper only and diacetyl only conditions.** (A) Worm tracks (n = 36) are plotted for a representative sensory integration assay of well-fed animals behaving in the presence of 50 mM CuSO$_4$ in water (blue stripe) with no attractant (location out of view to the right). Regions of the plate that were not able to be tracked are in gray with the edge of the plate indicated in black. Tracks are plotted and color coded for time. (B) Worm tracks (n = 40) are plotted for a representative sensory integration assay of 3 hour food-deprived animals. Conditions and plotting the same as in A. (C) Worm tracks (n = 33) are plotted for a representative sensory integration assay of well-fed animals behaving in the presence of no barrier (blue stripe) with attractant is 1 μL 0.2% diacetyl (1:500) in 100% ethanol (location not shown). Plotting the same as in A. (D) Worm tracks (n = 33) are plotted for a representative sensory integration assay of 3 hour food-deprived animals. Conditions the same as in C and plotting the same as in A.
(TIF)

**S4 Fig (to accompany Fig 2). Description of measurements to define tracking dynamics and additional treatment groups.** (A) Measuring Barrier Crossings. Worm tracks (n = 31) are plotted for the entire 45 minutes of a representative sensory integration assay. 26 tracks that terminate at the copper barrier are plotted in a unique color with the start of each track labelled with a numbered, circular marker. To obtain a measure of barrier crossing, the number of unique, continuous, reverse moving tracks (i.e. tracks that terminate on the right side of the copper barrier) was subtracted from the number of forward moving tracks (i.e. tracks that end on the left side of the copper barrier) and then divided by the estimated number of animals in the experiment. In the example experiment shown, 24 unique forward tracks and 2 reverse tracks were found for the 31 animals assayed, resulting in a Barrier Crossings score of 0.7097 for this experiment after 45 mins of recording. (B) Measuring Probability of Location. 9 unique animal tracks are plotted in a 3 mm x 3 mm field-of-view, a 9 mm$^2$ inset of a 45-minute example experiment. The midpoint positions of the animals at each frame are plotted as circles connected by lines. Midpoints located in Bin X (1 mm wide) are represented by filled circles while midpoints located in the neighboring bins (Bin X-1 and Bin X+1, each 1 mm wide) are represented by open circles. The probability of a worm being located in Bin X is calculated by dividing the number of tracked midpoints in Bin X by the total number of tracked points in all bins. In the small example area shown, there are 139 points in Bin X and a total of 345 points across all 3 bins resulting in a p(Location) score of 0.4029. In the entire field of view there are 45 bins, yielding an average p(Location) score of 0.0222. This analysis was used in Fig 2D, 2G and 2J. (C) Measuring Velocity. 10 seconds (i.e. 30 frames) of a single example worm track is plotted. The midpoint positions of the worm at each frame as identified by WormLab are plotted as

filled circles connected by lines. For each time $t$, the velocity was calculated by computing the Euclidean distance of the track from time $t-1$ second to time $t+1$ second and dividing by the length of time, 2 seconds. Because these videos lack the spatial resolution necessary to accurately estimate absolute path length (and thus body bends), Euclidean distance is used. In the example given, the Euclidean distance of the 2 second time window centered at time $t$ was 425.7 μm resulting in an instantaneous velocity of 212.9 μm/s. Velocity was calculated for every time point in this way and averaged across 1 mm bins for Fig 2E, 2H and 2K. (D) Graph of the chemotactic index (# animals on odor side / total # of animals) for sensory integration assays of well-fed animals behaving in the presence of 50mM $CuSO_4$ and 1 μL 0.2% diacetyl (1:500) over time (15, 30, 45 minutes). Well-fed (WF) animals appear with black dots and food-deprived (FD) animals are indicated with blue dots. Each dot represents a single plate of animals, with each plate measured at each time point (matched, n = 6). Analyzed using a Two-Way ANOVA, determined to have significant differences across well-fed and food-deprived conditions. WF/FD comparisons were then performed as pairwise comparisons within each time period as t-tests with Bonferroni's correction for multiple comparisons. * $p < 0.5$, ** $p < 0.01$, *** $p < 0.001$, **** $p < 0.0001$, ns $p > 0.05$. (E, F, G) The fraction of mean cumulative forward and reverse barrier crossings is plotted at three time points (15, 30, and 45 mins). Well-fed (WF) animals appear with black dots and food-deprived (FD) animals are indicated with blue dots. Each dot represents a single plate of animals. E) 50 mM $CuSO_4$ and 0.2% diacetyl F) 50 mM $CuSO_4$, no diacetyl G) No copper, 0.2% (1:500) diacetyl. Graphs are analyzed using a two-way ANOVA to determine significant differences across well-fed and food-deprived conditions. WF/FD comparisons were then performed as pairwise comparisons within each time period as t-tests with Bonferroni corrections for multiple comparisons. * $p < 0.5$, ** $p < 0.01$, *** $p < 0.001$, **** $p < 0.0001$, ns $p > 0.05$.
(TIF)

**S1 Table. All worm strains used in the experiments.** Strain ID, genotype/allele, and how it is referenced in the paper is provided. If the strain is first described here (all IV strains), the method of creation is provided. For strains in which chemotactic index was low, the average chemotactic index of WF and FD animals in the absence of copper (diacetyl alone) is provided.
(XLSX)

**S2 Table. All p-values for Fig 2D, 2G, 2J, 2E, 2H and 2K.** The p-values shown are the result of multiple unpaired t-tests with Welch's correction with Holm-Šídák post-hoc tests correction for multiple comparisons. Adjusted p-values are shown, with yellow shading for adjusted p-values $< 0.05$, same shading as in Fig 2D, 2G, 2J, 2E, 2H and 2K.
(XLSX)

**S1 Movie. Sensory integration behavior of well-fed animals.** ~150 Well-fed wildtype animals are placed in the standard sensory integration assay. Bracket indicates origin where animals are placed, spot shows position of 1:500 diacetyl odor, midline indicates repellent $CuSO_4$ barrier.
(MOV)

**S2 Movie. Sensory integration behavior of food-deprived animals.** ~150 Wildtype animals' food-deprived for three hours are placed in sensory integration behavior assay. Bracket indicates origin where animals are placed, spot shows position of 1:500 diacetyl odor, midline indicates $CuSO_4$ barrier.
(MOV)

**S3 Movie. HLH-30::GFP dynamics during food deprivation.** In this sample video, a maximum intensity projection of one *hlh-30::gfp* transgenic animal is shown from 45 mins to 225 mins post starvation, sped up to 3 fps.
(AVI)

## Acknowledgments

We thank A. Dillin, T. Ishihara, S. Lockery, A. Samuelson, E. Troemel, M. Zhen, the National BioResource Project (NBRP, Japan) and Caenorhabditis Genetics Center (CGC) for strains; C. Bargmann, E. Hallem, M. Hilliard, A. van der Linden, P. McGrath, D. Pilgrim and P. Sengupta for constructs; S. Srinivasan and lab members for RNAi clones and help with Oil-Red O staining; M. Tamés, Z. Liu, and C. Yang for technical help with behavioral and imaging studies; and Waitt Advanced Biophotonics Core. We are also grateful to J. Wang, and members of the Chalasani lab for critical comments, advice, and insights.

## Author Contributions

**Conceptualization:** Molly A. Matty, Hiu E. Lau, Sreekanth H. Chalasani.

**Data curation:** Molly A. Matty, Hiu E. Lau.

**Formal analysis:** Molly A. Matty, Hiu E. Lau, Jessica A. Haley, Anupama Singh, Ahana Chakraborty, Karina Kono.

**Funding acquisition:** Sreekanth H. Chalasani.

**Investigation:** Anupama Singh, Ahana Chakraborty, Karina Kono, Kirthi C. Reddy.

**Methodology:** Molly A. Matty, Hiu E. Lau, Jessica A. Haley.

**Project administration:** Sreekanth H. Chalasani.

**Resources:** Kirthi C. Reddy.

**Supervision:** Malene Hansen, Sreekanth H. Chalasani.

**Visualization:** Molly A. Matty.

**Writing – original draft:** Molly A. Matty, Hiu E. Lau, Sreekanth H. Chalasani.

**Writing – review & editing:** Molly A. Matty, Jessica A. Haley, Malene Hansen, Sreekanth H. Chalasani.

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
