## [Decision Letter · Decision Letter 0]

12 Oct 2021

Dear Dr %Chalasani%,

Thank you very much for submitting your Research Article entitled 'Intestine-to-neuronal signaling alters risk-taking behaviors in food-deprived Caenorhabditis elegans' to PLOS Genetics.

The manuscript was fully evaluated at the editorial level and by independent peer reviewers. The reviewers appreciated the attention to an important problem, but raised some substantial concerns about the current manuscript. Based on the reviews, we will not be able to accept this version of the manuscript, but we would be willing to review a much-revised version. We cannot, of course, promise publication at that time.

If you decide to revise the manuscript for further consideration at PLOS Genetics, please aim to resubmit within the next 60 days, unless it will take extra time to address the concerns of the reviewers, in which case we would appreciate an expected resubmission date by email to plosgenetics@plos.org.

[LINK]

We are sorry that we cannot be more positive about your manuscript at this stage. Please do not hesitate to contact us if you have any concerns or questions.

Yours sincerely,

Kaveh Ashrafi

Associate Editor

PLOS Genetics

Gregory P. Copenhaver

Editor-in-Chief

PLOS Genetics

Reviewer's Responses to Questions

**Comments to the Authors:**

Reviewer #1: In this manuscript, the authors studied molecular mechanisms underlying starvation-dependent decreases in avoidance responses toward copper ions using sensory integration assays between copper ions and attractive odorants and quantifying avoidance responses toward copper ions. They found that the mechanical stimuli were not sufficient for the decreased copper ion avoidance responses by analyzing the effects of Sephadex gel beads as tactile stimulation. Next, they used C. elegans mutants of a MondoA ortholog MML-1, a transcription factor regulated by glucose in the cytoplasm, and bHLH transcription factor HLH-30, which was reported to be activated by MML-1. They showed that mutants of MML-1 and HLH-30 showed defects in starvation-dependent decreased copper ion responses. Furthermore, nuclear localization of MML-1 and HLH-30 was altered by the presence of food. On the other hand, mutants of multiple insulin-like genes which were reported to have HLH-30 binding elements, the insulin/IGF-I receptor homolog DAF-2, PI 3-kinase AGE-1 and a TORC2 complex component RICT-1 also showed defects in starvation-dependent decrease in copper ion responses. Using cell-type-specific rescue experiments, they showed that daf-2 and rict-1 functioned in ASI neurons and mml-1 functioned in the intestine in decreased copper ion responses in starvation conditions. From these experimental results, they proposed a model in which MML-1 and HLH-30 in the cytoplasm promote insulin-like peptide release from the intestine in food-deprived conditions, which in turn activates DAF-2/RICT-1 signaling in the ASI neuron to decrease responses toward copper ions.

The manuscript is clearly described and the proposed model is interesting. It is noteworthy that the authors found a new function for mml-1 in behavioral plasticity. However, there are several issues that should be addressed before publication.

Major points

1. Although the authors mentioned about some previous researches reporting starvation-dependent behavioral changes (line 75-79, page 4), they did not mention about a previous research, J. Vis. Exp. (125), e55939, doi:10.3791/55939 (2017), showing that avoidance of copper ions was decreased in interaction assay between copper ions and food, which is a similar phenomenon shown in Fig. 1A, B. This paper should be referenced and the authors could revise the manuscript to clearly show the differences from the past findings.

2. In a paper, EMBO J 2011 Mar 16;30(6):1110-22, which was referenced, it was shown that avoidance behavior and ASH activities in response to copper ions were decreased after food deprivation. Therefore, a conclusion, “food-deprived animals display reduced sensitivity to copper” (line141-142, page 7), is not a novel finding. Rather, in the current paper, the authors observed only the behavioral responses to copper without seeing neuronal response to copper. So, the current paper does not show experimental evidence showing sensitivity to copper is altered by food deprivation.

3. In line 212-213, page 10, the authors conclude that “the lack of food in the C. elegans intestine, but not the absence of chemosensory cues, reduces the animals’ sensitivity to copper”. In the experiment in Fig. 3E, F, the authors examined the tactile stimuli but not the chemosensory stimuli.

4. As mentioned in line 171-172, page 8, the trial number shown in Fig. 2F is smaller than that in Fig. 2C or I. In view of Fig. 1E where chemotaxis with only CuSO4 significantly changes, the lack of significant difference in Fig. 2F is formally a contradiction. To conclude the behavioral mechanisms presented, the authors should repeat the tracking experiment observing copper ion responses on agar plates with only CuSO4 (Fig. 2F, G, H).

5. The authors concluded that “MML-1 and HLH-30 accumulation in the cytoplasm (in food deprived animals) enables the intestine to release peptide(s) relaying a “lack of glucose” signal to other tissues” (line 342-344, page 15-16). But the experimental results about HLH-30 localization (Fig. 4G) are difficult to interpret and are not sufficient to support the above conclusion. First, 1hr food deprivation decreased cytoplasmic HLH-30 accumulation (Fig. 4G), whereas it caused decreased copper avoidance (Fig. 1B). The authors should discuss HLH-30 function after 1hr food deprivation. Second, 3hr food deprivation increased cytoplasmic HLH-30 accumulation compared to that after 1hr food deprivation. The authors should also compare HLH-30 localization between well-fed and 3hr FD animals to judge if 3hr FD increases cytoplasmic accumulation of HLH-30.

6. To assess whether mml-1 and daf-2 function in the same pathway, the authors examined the phenotype of the double mutant of mml-1 and daf-2 and compared it with each single mutant (line 288-290, page 13; Fig. 5E). Because in both mml-1 and daf-2 mutants there is no significant difference between WF and FD animals in copper response (Fig. 5E), authors cannot claim that these are on the same pathway just by looking at double mutants. Assuming daf-2 and mml-1 are on different pathways, what would be the double mutant phenotype? Probably the same. Phenotypes of single mutants already show that both MML-1 and DAF-2 need to be intact for the starvation response to occur. In line 288, authors state "upstream MML-1 and downstream DAF-2", but this is not a genetically proven relationship, either. Therefore, if the authors want to claim more on the relationship between mml-1/hlh-30 and insulin pathway, they should test it in other ways. For example, fluorescence imaging experiment might be possible. As the authors suggested that expression of some insulin-like genes may be regulated by HLH-30, the authors could test insulin-like peptide expression using fluorescence reporter in several conditions, such as food deprivation and genetic background of mml-1 and hlh-30 mutants.

Minor points

1. Regarding the daf-28(sa191) mutant, the authors mentioned that this mutant is defective in multiple insulin-like peptides based on the previous findings (line265-267, page 12). This mutant was also shown to be defective in daf-7, TGFb, function (PLoS Genet. 2016 Dec 7;12(12):e1006450.). The authors could discuss about a possible function of signaling pathway other than insulin pathway, such as TGFb pathway, in copper response plasticity.

2. In Supplemental table, three lines of rict-1 transgenic lines carrying Pstr-3::rict-1 are shown. But Fig. 6C shows results of only two of those transgenic lines.

3. The authors show the possibility that PI3K and Rictor (RICT-1) may function in the same pathway in ASI neurons (line 310-311, page 14). To examine the genetic interactions between mTORC2 pathway and DAF-2/PI3K pathway, the authors could use sgk-1 mutants. SGK-1 was reported to function downstream of RICT-1 and there are a loss-of-function mutant, sgk-1(ok538) and a gain-of-function mutant, sgk-1(ft15), which are useful for epistasis analysis (Genes Dev 2016 Jul 1;30(13):1515-28).

Reviewer #2: Chalasani and colleagues report a study of altered risk-taking behavior in response to food deprivation, establishing a novel assay combining food deprivation with subsequent challenge to toxic copper and a food signal. Genetic analysis of established mutants and pathways are conducted to support a narrative involving the effect of food deprivation in modulating the transcriptional regulation of insulin ligands in the intestine, acting on the DAF-2 insulin receptor pathway in the ASI chemosensory neurons to modulate behavioral output. The study of gut-to-brain signaling is an area of wide interest, and the C. elegans system is well-suited to mechanistic dissection of molecular and cellular mechanisms and circuits involved in a simple animal system with tractable behavioral outputs. The manuscript is ambitious in tackling this question and touches upon all of the interesting relevant questions that might connect the observed behavioral system to the underlying molecular signaling.

Strengths of the manuscript include the careful analysis of behavior and the manner in which the authors address how to interpret the design of their behavioral assay, including control experiments that get at potential caveats and limitations. Important weakness in the manuscript that should be addressed are in the somewhat superficial genetic analysis. Behavioral genetics is challenging because the effect sizes are often not large, and thus appropriate rigor is needed if there is to be any sort of take-home message from the analysis.

Some suggestions:

1. The authors conclude that food deprivation and not absence of chemosensory cues, account for the alterations during food deprivation. Beads are tested. What about heat-killed and/or aztreonam-treated bacteria?

2. Mml-1 effect is not a large one. Rescue experiments and/or multiple alleles should be examined.

3. On the other hand, the hlh-30 shows a robust effect. As the story builds based on the rationale that HLH-30 may regulate insulin gene expression, rescue experiments and intestine-specific rescue in particular would strengthen the intestine-behavior link.\\

4. Are the insulin ligands expressed in the intestine?

5. In the analysis of the insulin pathway (ligands and pathway components), even where statistical analysis supports the authors’ interpretation, there is a trend towards negative results. Particularly where conclusions are drawn, analysis of multiple alleles and rescue experiments are necessary—i.e. for ins-23, ins-31. For daf-28, lf alleles should probably be examined. The interpretation of the sa191 allele is complex (see Kulalert et al., 2013).

6. The genetic analysis of daf-2 is confusing (admittedly, it's always confusing). In one set of rescue experiments, the m596 allele is used, but ASI rescue is not done. In another set of experiments, all analysis is done in the mml-1; daf-2 (e1370) double mutant; each single has a phenotype. But instead of simply showing single mutant rescue, a somewhat convoluted double mutant rescue using tissue-specific promoters is shown. This should be clarified with corresponding rescue of single mutants. Also, the switching of alleles in the case of daf-2 is non-trivial given the many alleles of daf-2 and their altered strengths and activities.

7. Rict-1 analysis shows small effects; multiple alleles and rescue not only with ASI promoter but also intestine and pan-neuronal promoters should be shown.\\

The paper is ambitious in scope, and there is a lot of data in this paper. Where the essential basic amount of rigor cannot be provided, perhaps simply omitting these data/claims would also be a reasonable option.

Reviewer #3: Matty et al. explore the modulation of chemosensory repulsion by feeding state through an impressive array of experiments that explore behavioral, sensory, and molecular dimensions of the phenomenon. The experiments are done to a high standard for the most part. However, the manuscript is a little confusing, and the broader conceptual framework needs to be reconsidered in my opinion.

Overall comments.

I question that the phenomenon under study here really has much to do with “integration”—which implies to me an interaction or dependency between two kinds of stimuli or information. Here, it is clear that starvation modulates response to the repellent copper, but irrespective of whether there is an attractive odor present. Sometimes “integration” seems to refer to the DA / Cu decision-making task (“sensory integration assay”), although these modalities seem to be independent of one another in their results, and sometimes to the effect of intestinal peptides on ASI. It’s not clear to me how a uni-modal interaction between the intestine and ASI constitutes “integration,” particularly in cases like line 98, where “integrate” is simply used as a transitive verb on a single object (“peptides”). I can’t see a cell responding to extracellular cues as integration, necessarily…it’s just responding to a signal. So this framework feels muddled… the authors need to be clearer about what they think “integration” means in the context of their experiments and be clear about which of the phenomena they describe constitute its mechanism. Their findings would be greatly strengthened by demonstrating an effect of starvation (and better yet, the putative peptide signals from the intestine in the absence of real starvation) on ASI neuronal function and/or circuit activity related to behavioral avoidance.

Specific comments

1. Previous research implicates ASH in being responsible for aversive responses to Cu, and inhibitory signaling between ASI and ASH. I’m unaware of studies showing that ASI plays a role in Cu avoidance behavior… where experiments done excluding ASH involvement? What led the authors to ASI? What circuit mechanisms plausibly explains ASI’s involvement?

2. Ghosh et al. showed that DA will induce animals to cross a hyperosmotic barrier, but no compelling effect is seen in Fig 1C – is this related to the low concentration of DA used here? A much higher concentration is used in Fig 1G.

2. Figure 2C – the results here are non-significant, but it is doubtful that a significant differencts of the size shown could be detected given the sample size used. I don’t think any conclusion should be drawn form this, or the experiment should be repeated with samples appropriate to the effect size of interest.

3. Fig 5E and Fig 6C - last two groups in both have identically labelled x-axis categories

4. Fig 5. Are age-1 animals sickly or movement impaired? Difficult to conclude anything about the effects of FD if they are generally poor at chemotaxis.

**Have all data underlying the figures and results presented in the manuscript been provided?**

Reviewer #1: Yes

Reviewer #2: Yes

Reviewer #3: Yes

PLOS authors have the option to publish the peer review history of their article (what does this mean?). If published, this will include your full peer review and any attached files.

Reviewer #1: No

Reviewer #2: No

Reviewer #3: No

---

## [Decision Letter · Decision Letter 1]

23 Mar 2022

Dear Dr. Chalasani,

Thank you very much for submitting your Research Article entitled 'Intestine-to-neuronal signaling alters risk-taking behaviors in food-deprived Caenorhabditis elegans' to PLOS Genetics.

The manuscript was fully evaluated at the editorial level and by independent peer reviewers. The reviewers appreciated the attention to an important topic but identified some concerns that we ask you address in a revised manuscript. Specifically, the following point raised by Reviewer #2 remains significant: "The narrative argues strongly for a role for MML-1 and HLH-30 in the intestine. Rescue experiments and/or multiple alleles represent a minimum level of genetic rigor to make the point."  We recognize that your attempt at intestinal rescue of *hlh-30* has not been successful.  It is possible for the identified insulins to play a role in this behavior without being directly regulated by either MML-1 or HLH-30.  Therefore, we ask you to either address this point experimentally or modify the text to clearly indicate this ambiguity. 

We would also appreciate your comments in addressing the other two points of the same reviewer.  These comments need not be included in the text of the manuscript.

[LINK]

Yours sincerely,

Kaveh Ashrafi

Associate Editor

PLOS Genetics

Gregory P. Copenhaver

Editor-in-Chief

PLOS Genetics

Reviewer's Responses to Questions

**Comments to the Authors:**

Reviewer #1: Upon revision, authors satisfactorily addressed the questions raised and this reviewer has no remaining concern.

Reviewer #2: The revised manuscript has been improved by the removal of data that are not as well substantiated. The narrative of intestine-to-neuron signaling would benefit from strengthening the rigor of the genetic analysis, as suggested in the initial review. I repeat basic suggestions from my initial review:

1) Heat-killed and/or aztreonam-treated bacteria can not only assess for the effect of volatile cues, but also water-soluble cues, which are not assayed in the revision.

2) The narrative argues strongly for a role for MML-1 and HLH-30 in the intestine. Rescue experiments and/or multiple alleles represent a minimum level of genetic rigor to make the point.

3) The ins-31 intestine rescue experiments have been performed, but the data in Figure 5D are not convincing. Why does the mutant data look so different from the data with the presumably nonfunctional neuronally-expressed ins-31?

Reviewer #3: All of my concerns have been addressed

**Have all data underlying the figures and results presented in the manuscript been provided?**

Reviewer #1: Yes

Reviewer #2: Yes

Reviewer #3: Yes

PLOS authors have the option to publish the peer review history of their article (what does this mean?). If published, this will include your full peer review and any attached files.

Reviewer #1: No

Reviewer #2: No

Reviewer #3: No

---

## [Editor Report · Decision Letter 2]

30 Mar 2022

Dear Dr %Chalasani%,

We are pleased to inform you that your manuscript entitled "Intestine-to-neuronal signaling alters risk-taking behaviors in food-deprived Caenorhabditis elegans" has been editorially accepted for publication in PLOS Genetics. Congratulations!

Yours sincerely,

Kaveh Ashrafi

Associate Editor

PLOS Genetics

Gregory P. Copenhaver

Editor-in-Chief

PLOS Genetics

Comments from the reviewers (if applicable):

**Data Deposition**

http://datadryad.org/submit?journalID=pgenetics&manu=PGENETICS-D-21-01159R2

**Press Queries**

---

## [Editor Report · Acceptance letter]

13 Apr 2022

PGENETICS-D-21-01159R2 

Intestine-to-neuronal signaling alters risk-taking behaviors in food-deprived Caenorhabditis elegans 

Dear Dr Chalasani, 

We are pleased to inform you that your manuscript entitled "Intestine-to-neuronal signaling alters risk-taking behaviors in food-deprived Caenorhabditis elegans" has been formally accepted for publication in PLOS Genetics! Your manuscript is now with our production department and you will be notified of the publication date in due course.

With kind regards,

Agnes Pap

PLOS Genetics

On behalf of:
